# Epoxy and Polyester Composites’ Characteristics under Tribological Loading Conditions

**DOI:** 10.3390/polym13142230

**Published:** 2021-07-07

**Authors:** Jasem Ghanem Alotaibi, Ayedh Eid Alajmi, Gabrel A. Mehoub, Belal F. Yousif

**Affiliations:** 1Department of Automotive and Marine Engineering Technology, Public Authority for Applied Education and Training, Adailiyah 42325, Kuwait; jg.alotaibi@paaet.edu.kw (J.G.A.); ae.alajmi@paaet.edu.kw (A.E.A.); 2Mechanical Technology Department, High Institute of Science and Technology, Qasr Bin Ghashir 22131, Libya; GMehoub@hinstitute-bcv.edu.ly; 3Faculty of Health, Engineering and Sciences, University of Southern Queensland, Toowoomba, QLD 4350, Australia

**Keywords:** composites, friction and wear behaviour, tribological properties, reinforcement

## Abstract

This research examines the friction and dry wear behaviours of glass fibre-reinforced epoxy (GFRE) and glass fibre-reinforced polyester (GFRP) composites. Three fibre orientations—parallel orientation (P–O), anti-parallel orientation (AP–O), and normal orientation (N–O)—and various sliding distances from 0–15 km were examined. The experiments were carried out using a block-on-ring configuration at room temperature, an applied load of 30 N, and a sliding velocity of 2.8 m/s. During the sliding, interface temperatures and frictional forces were captured and recorded. Worn surfaces were examined using scanning electron microscopy to identify the damage. The highest wear rates for GFRE composites occurred in those with AP–O fibres, while the highest wear rates for GFRP composites occurred in those with P–O fibres. At longer sliding distances, composites with P–O and N–O fibres had the lowest wear rates. The highest friction coefficient was observed for composites with N–O and P–O fibres at higher sliding speeds. The lowest friction coefficient value (0.25) was for composites with AP–O fibres. GFRP composites with P–O fibres had a higher wear rate than those with N–O fibres at the maximum speed.

## 1. Introduction

Given the rapid global developments and challenges associated with the use of metals in tribological industrial applications, the tribological behaviours of polymeric composites is attracting increased research attention. Fibre-reinforced polymeric composites have numerous mechanical advantages over metal materials, including higher specific strength, lower weight and lower raw material and processing costs. Composite materials have been used in many applications with superior results, including the production of structural materials in the aerospace industry [1,2]. Thus, the tribological properties of polymeric composites are a key topic of interest for researchers. Various characteristics of composites have been studied, including their friction and wear performance in brakes, clutches, and nuts and bolts [3,4]. Shalwan and Yousif [5] explain that friction is the energy dissipated at the material’s contact surface. Wear, meaning the removal of a solid surface, can be measured in terms of weight loss, wear resistance, or specific wear rate (SWR) [4]. Friction and wear have major effects on the efficient operation and life span of industrial machinery [6], and are the most common problems encountered in industrial engineering, necessitating the replacement of machine components and assemblies [7,8]. Consequently, polymer materials are being increasingly adopted in industrialised countries. Thus, there is a clear need to understand the tribological behaviours of polymer composites [9,10].

The few beneficial applications of friction include tyres, brakes, clutches, and nuts and bolts, while unproductive friction and wear occurs in gas turbines, cams and bearings, and external combustion engines [11]. Friction involves three basic elements: (i) type and strength of interfacial bonds, (ii) shearing and rupture of rubbing materials around the contact area, and (iii) contact area [9]. Factors affecting the deformation and adhesion components of friction include sliding velocity, applied load, and temperature [12]. Friction can damage material surfaces and change the mechanical properties of composites. Finally, friction results in the conversion of mechanical energy to heat, leading to the deformation of materials.

Researchers have examined the tribological performance of polymeric composites reinforced with synthetic fibres such as glass [1] and carbon [13]. However, an understanding of the tribological behaviours of fibre-reinforced thermoset composites is still lacking. Shalwan and Yousif [5] have called for further studies to identify the wear and friction characteristics of glass fibre-reinforced thermoset composites. This has motivated the current study, which compares the wear and friction behaviours of glass fibre-reinforced epoxy (GFRE) and glass fibre-reinforced polyester (GFRP) composites with three different fibre orientations at different sliding distances (0–15 km). In this work, the operating parameters were limited to 30 N of applied load and 2.8 m/s of sliding velocity. This is mainly due to the limitation of thermoset performance at severe tribological loading conditions (above 50 N applied load and higher than 2.8 m/s sliding velocity), [3,14]. In the literature, at a high applied load of 50 N and sliding velocity of 3.9 m/s, the surface of the thermosets dramatically softened and high removal of material have been reported leading to decomposition of the composites.

In light of the above, the need to understand the influence of the fibre orientations on the tribological performance of both epoxy and polyester motivated the current study. The main aim of the work is to identify the optimum fibre orientation in both composites to yield the optimum performance of the composite under tribological loading conditions. Furthermore, this will assist industries and researchers in implementing such composites in tribological applications and give further understanding on the critical impact of the fibre orientation on the performance of such composites.

## 2. Material Preparation and Experimental Procedure

### 2.1. Material Preparation

Given that the aim of this study was to investigate surface damage under different sliding conditions, materials that could easily display surface damage were required. Well-known materials include neat epoxy (NE), and combinations of materials were used to fulfil the technical conditions.

Epoxy resins range from high melting point solids to viscous liquids and have numerous mechanical properties, including <2% shrinkage, high hardness, and high chemical resistance. Liquid epoxy resin is used for a range of purposes, including automotive parts and casting, and is highly resistant to adhesives and alkalis. Thus, liquid epoxy resin (DER 331, supplied by Clearbox, Clearbox Systems Pty Ltd., Macquarie Park, NSW, Australia) was mixed with hardener in a 2:1 ratio. The mixture was melted in a mould and placed in a vacuum station (MCP 004 PLC, Multistation, Dinard, France) at room temperature for 24 h to eliminate air bubbles.

GFRE and GFRP were selected as the reinforcement materials because of their high resistance to chemicals and other environmental factors. Similar techniques to those for the epoxy resin were used to fabricate the synthetic specimens. Three fibre orientations were considered in the tests. To ensure superior properties, the specimens had a specific fibre volume (vf) of 48%. Figure 1 shows micrographs of the original surfaces of both composite materials along with their fibre orientations.

GFRP can exhibit different lengths, widths, and weights. In the specimen used in this study, fibre length was 20–30 mm, and fibre mass was 450 g/m^2^, fibre diameter is 10–30 µm. Unsaturated polyester (Revesol P9509) was added to methylethylketone peroxide at ambient room temperature. Both reinforcement and polyester materials were supplied by Kong Tat Company (Bandar Indahpura, Malaysia). The composite samples were 30 × 20 × 20 mm in size, with different fibre orientations (see Figure 2).

### 2.2. Experimental Set Up and Procedure

A block-on-ring machine was used to conduct the experiments. Specimen surfaces (10 × 10 × 20 mm) were tested against a stainless steel (AISI 304, hardness = 1250 HB, Ra = 0.1 µm). Prior to each test, the counterface was smoothed using Sic G2000, then cleaned with an acetone-moistened cloth. Figure 3 shows the block-on-ring configuration, with the load cell, samples, counterface, and sample holder. The load cell was connected to the computer to capture the frictional forces during the experiments.

The contact between the two surfaces is cylindrical against flat and the maximum Hertzian contact pressure (MPa) is determined to be 3.2, 1.6, 10.5, and 6.8 for neat epoxy, neat polyester, GFRE, and GRFP, respectively. The roughness of the wear track was gauged before and after the experiment using a Mahr Perthometer S2 (Mahr Federal Inc., Providence, RI, USA). Due to the close contact between the stainless steel and the specimen, abrasive paper (Sic G2000) and a dry, soft brush, respectively, were used to polish and clean the specimen contact surfaces. The composite surfaces varied in terms of roughness. For instance, the average roughness of composites with normal orientation (N–O) fibres was 0.70 µm, while the average roughness of composites with parallel orientation (P–O) and anti-parallel orientation (AP–O) fibres was 0.30 µm.

### 2.3. Experimental Procedure

Experiments were conducted at room temperature (28 °C) using a constant applied load of 30 N, a sliding velocity of 2.8 m/s, and a sliding distance of 0–15 km. A new specimen was used for each sliding distance. Before and after each test, a dry soft brush was used to clean the prepared composite specimen. To measure weight loss, a Setra weight balance (±0.1 mg) was used to determine the weights of composite specimens before and after each test. Each set of tests has been repeated three times for three different samples. A scanning electron microscope (JEOL) was used to examine the morphology of the composite surfaces. A thermal imaging camera was used to determine the interface temperatures before and at certain periods during each test and to show heat allocation following the test. SWR was measured to explore the damage to the specimen surface caused by wear using Equation (1):(1)$$SWR=\frac{\frac{\Delta W}{\rho }}{L}\times D$$
where:*SWR*: specific wear rateΔW:weight differenceρ:density of the sample*L*: applied load

Therefore, to estimate SWR, theoretical rules were applied to measure the relationship between sliding distance and specimen weight before and after each test. Friction forces were obtained using tribology software, which was connected to the block-on-ring machine. Hence, shear force readings were automatically registered according to the generated data. Each tribological test was repeated several times, and the average values were recorded.

The experimental setup includes an infrared thermometer (Extech 42580, Test Equipment Depot, Melrose, MA, USA) to deduct the interface temperature between the surface and the counterface. The calibration and the measurement procedure was detailed in [15]. In summary, a thermocouple was placed between the composite and the counterface surfaces in a stationary position. The counterface was heated using an external heat gun and then the interface temperature was captured by the thermocouple and the infrared simultaneously. A calibration equation was developed and used to determine the real interface temperature while using the infrared thermometer.

## 3. Results and Discussion

### 3.1. Wear Behaviour

To study the wear behaviours of NE, neat polyester (NP), GFRE, and GFRP, a series of experiments were conducted at different operating parameters and fibre orientations (N–O, P–O and AP–O). Figure 4a shows the results for the SWR of NE and GFRE of different fibre orientations. Given the extremely low SWRs of all selected materials, all values were multiplied by 1,000,000 (i.e., E6). The SWR of NE was comparatively higher than that of the epoxy composites; thus, these values are shown on the right-hand vertical axis using a different scale. Figure 4a shows that NE exhibited a higher SWR, reaching a steady state after approximately 5 km. In contrast, epoxy composites showed a lower SWR compared with NE for all fibre orientations, reaching a steady state after approximately 10 km because it took longer for interactions to occur between the surface asperities. A further explanation of the roughness profile is given in the next section.

AP–O composites had a higher SWR than P–O or N–O composites, which exhibited a lower SWR after 5 km. However, the SWR of the AP-O composite was around 30% lower than that of NE, while the SWR of P–O and N–O composites was around 20% lower than that of NE. This may be explained by the removal of the fractured glass fibres from the proportionally harder phase (CSM). At an applied load of 30 N, the weight loss of composites increased significantly with increasing sliding velocity and sliding distance.

Figure 4b shows the results for the SWR of NP and GFRP composites of different fibre orientations. Given the extremely low SWR values, all obtained values were multiplied by 1,000,000. The highest SWR value occurred in the P–O composite at a sliding distance of approximately 3 km (given the time needed for the interaction between the surfaces). The lowest SWR values occurred in the AP–O and N–O composites at approximately 3 km, reaching a steady state after 6 km. After approximately 6 km, all composites reached a steady state, after which there were no marked differences in wear rates. N–O composites had the lowest SWR at all sliding distances tested, thus displaying superior wear performance. The SWR values of N-O composites clearly decreased as a result of the better wear behaviours. This may be attributable to the reinforcement arising from the adhesion between the glass fibres and the polyester resin. From a mechanical point of view, interface adhesion is enhanced by the mechanical characteristics of the composite. Thus, material strength is another possible reason for lower weight loss.

Table 1 summarises the SWRs of selected materials after reaching a steady state at 10 km. The optimum SWR for both GFRE and GFRP occurred in the N–O composites, which may be related to their mechanical properties in terms of interfacial adhesion and strength, lowering the hardness of the composite surface. For composites with AP–O fibres, the SWR of GFRP was higher than that of GFRE, while there was a marked difference between composites with P–O fibres. NE and NP had higher SWR values than either of the composites.

### 3.2. Friction Coefficients

Figure 5a shows the friction coefficient values for NE and GFRE composites at an applied load of 30 N and a sliding velocity of 2.8 m/s. In general, the friction coefficients of NE decreased slightly with increased sliding distance. The friction coefficients of N-O and P-O composites were similar, increasing at around 6 km before reaching a steady state. In contrast, AP–O composites had the lowest friction coefficient (0.25–0.3), around 29% less than that of NE. The different fibre orientations of GFRE exhibited different behaviours, ranging from 0.29 to 0.45, while the friction coefficient of NE was above 0.49.

Figure 5b shows the friction coefficient values for NP and GFRP composites at an applied load of 30 N and a sliding velocity of 2.8 m/s. For the composites in general, the friction coefficients increased initially, then began to decrease at a sliding distance of approximately 5 km. The GFRP composites exhibited friction coefficients of 0.2–0.3. However, the N–O and AP–O composites had the lowest friction coefficients of 0.23 and 0.28, respectively. The highest friction coefficient (0.42) occurred in NP. Sliding distance had no significant effect on the friction coefficients, and a steady state was not reached at any sliding distance. Nevertheless, friction coefficients reduced as sliding distance increased. This may be attributable to the strong transfer of film on the counterface and the existence of fibres and polyester. Longer sliding distances may impair the adhesion and associated interactions between the two sliding surfaces. The influence of the friction coefficient and wear performance may be illustrated by micrographs of the worn surfaces of the composites.

Figure 6 summarises the friction coefficients of all materials. The neat composites exhibited the highest friction coefficients at 10 km. Friction coefficients were also high for composites of different fibre orientations. However, the presence of fibres and strength of the interfacial adhesion prevents breakage and bending. Composites with AP–O fibres exhibited the lowest friction values. The friction value for GFRE with N–O fibres was higher than that of GFRP with N–O fibres. Of all fibre orientations for both composites, with AP–O fibres achieved the lowest friction value.

### 3.3. Interface Temperature

Figure 7a shows the influence of applied load and sliding distance on the interface temperature of GFRE composite with different fibre orientations (N–O, P–O, and AP–O) compared with NE. As friction, applied load, and sliding distance were maximised, the interface temperature was expected to increase. At an applied load of 30 N, the interface temperature gradually increased as sliding distance increased until approximately 5 km. The highest temperature recorded was 50 °C after 14 km for NE. AP–O and N–O composites had no significant effect on temperature, in contrast with the P–O composite, which reached 47 °C after 12 km. Previous experiments have shown that increased temperature has a long-term effect on composites. Results were obtained using a thermal imaging camera at 30 N applied load and 2.8 m/s sliding velocity. For the GFRE composite with N–O fibres, there was no change in temperature with an increase in sliding distance from 10 km to 14 km.

Figure 7b shows the maximum interface temperature, which occurred at a sliding distance of 15 km. The higher interface temperature was attributable to the high friction coefficients of NP and GFRP with P–O fibres compared with the NE composite. GFRP with AP–O fibres had a lower interface temperature compared with GFRE with AP–O fibres. Generally, sliding distance was closely related to interface temperature, which began to increase after 10 km. The thermal imaging camera was used in each test to provide further results.

### 3.4. Composite Surface Observation

This section discusses the results for the roughness profiles Ra. It should be mentioned here that the roughness profile was determined in the normal direction to the sliding direction. Results are recorded for both with and against the direction of the counterface. Figure 8 shows examples of the roughness profiles of GFRE and GFRP at different operating parameters.

Figure 9 summarises the roughness values of selected materials under an applied load of 30 N and a sliding distance of 15 km. Figure 9a shows the roughness values for GFRE. The highest value was recorded for AP–O in the direction of the counterface, while the lowest value was recorded for P–O. Figure 9b shows the roughness values of the GFRP under an applied load of 30 N and a sliding distance of 15 km. Compared with GFRE, the roughness value of GFRP with AP–O fibres was significantly lower in both directions. Moreover, GFRP with P–O fibres had a lower roughness value in the direction of the counterface. With respect to the surface roughness profile of NP and GFRP, the N–O composite had the highest roughness value at 3.536 m, while the AP–O had the lowest value. Additionally, the roughness profile of NP was slightly lower than that of NE.

In both composites, neat resins (Epoxy and Polyester) showed poor wear resistance, (Table 1). With regard to the composites, when the fibre ends were exposed to the rubbing area in normal orientation (N–O), both composites performed better than in the other orientations (P–O and AP–O). The micrographs of the composites’ surfaces in N–O are presented in Figure 10 and Figure 11. The worn surface of the epoxy composites showed no remarkable damage on the surfaces (Figure 10a,b) except a slight debonding associated with micro-cracks around the fibres. Figure 10c. There is a clear deformation on the surface, which could be due to film generation on the surface. There is no clear presence of fibre ends, which indicates that there is film generation on the composites surface. This may explain the smoothness of the surface when the roughness was measured and reported in Figure 9. Furthermore, the deformation on the surface represents the high resistance of the surface for material removal owing to low specific wear rate, Table 1. On the other hand, the worn surface of the epoxy composites showed dramatic deterioration on the surface (Figure 11a) since there was fibre breakage, debonding, delamination, and pull out fibres, as can be seen in Figure 11a,b. This leads to high roughness (Figure 9) and high specific wear rate, Table 1. Both Figure 10 and Figure 11 support the experimental results and the findings.

The results reported in Figure 5 showed that the epoxy composites performed better than the polyester composites at all fibre orientations. This could be due to many factors. Firstly, epoxy resin is much harder than polyester, and under tribological loading conditions the wear performance improves with the higher hardness of the surface, [16]. On the other hand, the interaction between the glass fibres and the resin may differ based on the type of resin. Epoxy is well known to have very high adhesion properties compared to polyester, [17]. This may suggest that the interfacial adhesion between the fibres and the resin may influence the material removal from the surface of the composite during the rubbing process. By comparing Figure 10 and Figure 11, it seems that the epoxy regions on the worn surface were adhering well with the fibres and the ends of the fibres carried the load during the sliding. However, Figure 11b showed broken fibres and deterioration in the polyester region near the fibres. This may explain the better performance of epoxy composites compared to the polyester ones.

### 3.5. Discussion and Comparison with Previous Published Works

In this section, the experimental results for NE, NP, GFRE, and GFRP with three fibre orientations (N–O, P–O and AP–O) are compared with results of other studies in terms of weight loss and frictional behaviour at various operating parameters. Figure 12 shows the results of several studies that have explored the SWR and frictional behaviour of composites. Shi et al. (2003) found that NE composites had the highest SWR [12]. Further, NP has been found to have lower SWR and friction coefficients, at 0.03234 and 0.23, respectively [13]. Pihtili (2009) found that GFRE had the lowest SWR and the highest friction coefficient [1]. The friction coefficients and SWR values of GFRP composites are shown in Figure 12. The same friction coefficient was found by Shalwan and Yousif (2012) [3]. However, the GFRP exhibited a low SWR (0.02).

## 4. Conclusions

The conclusions of this experiment are as follows:

The presence of fibres and fibre orientation have a significant influence on the wear rate and frictional behaviour of polymeric composites. GFRE composites with N–O fibres display improved wear and frictional behaviours.

Fibre orientation is highly influential in friction and wear performance. However, the SWR of AP–O composites was consistently higher than that of N–O composites. Moreover, sliding distance and applied load had little effect on tribological properties.

The dominant wear mechanisms were the detachment and breakage of fibres. The main wear mechanism for N–O and P–O composites was the formation of microcracks at the ends of fibres.

The worn surfaces of composites showed different wear mechanisms. For GFRP, plastic deformation and softening occurred during sliding, deteriorating the surface. For GFRE, there was less damage on the surface, despite the presence of microcracks, indicating the high wear resistance in the interface.

Increasing temperature and frictional force substantially affected the tribological properties of both polymeric composites. However, interface temperature has a pronounced influence on frictional force and material removal from the composite surface. Moreover, there were different appearances for the wear mechanism depending on the heat generated by the frictional force.

## Figures and Tables

**Figure 1 polymers-13-02230-f001:**
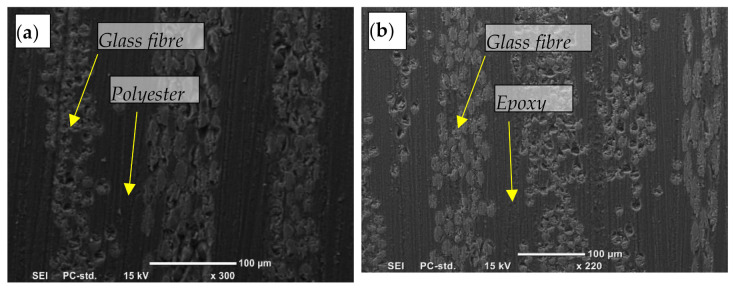
Micrographs of the original surfaces of (**a**) GFRP and (**b**) GFRE.

**Figure 2 polymers-13-02230-f002:**
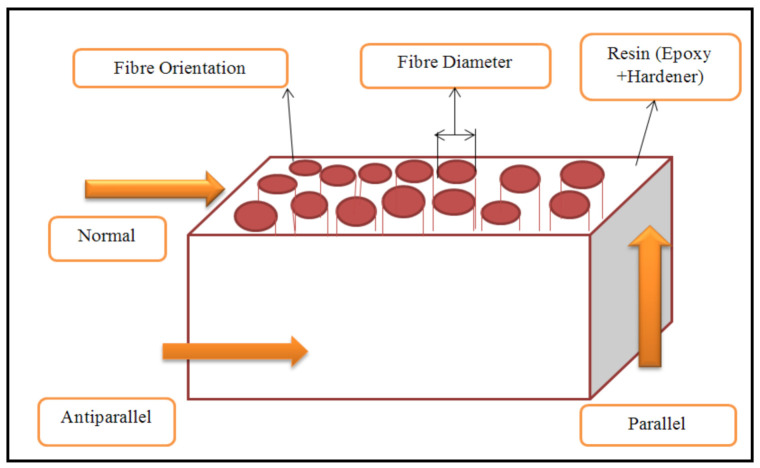
Schematic showing the orientation of fibres with respect to sliding direction.

**Figure 3 polymers-13-02230-f003:**
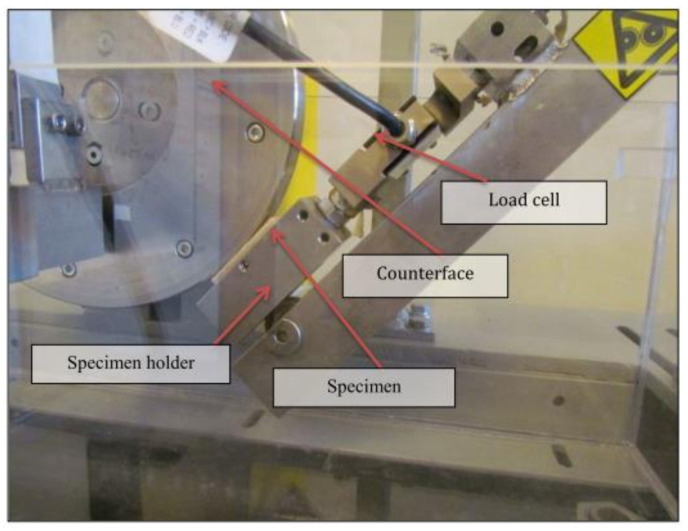
Block-on-ring machine configuration [11].

**Figure 4 polymers-13-02230-f004:**
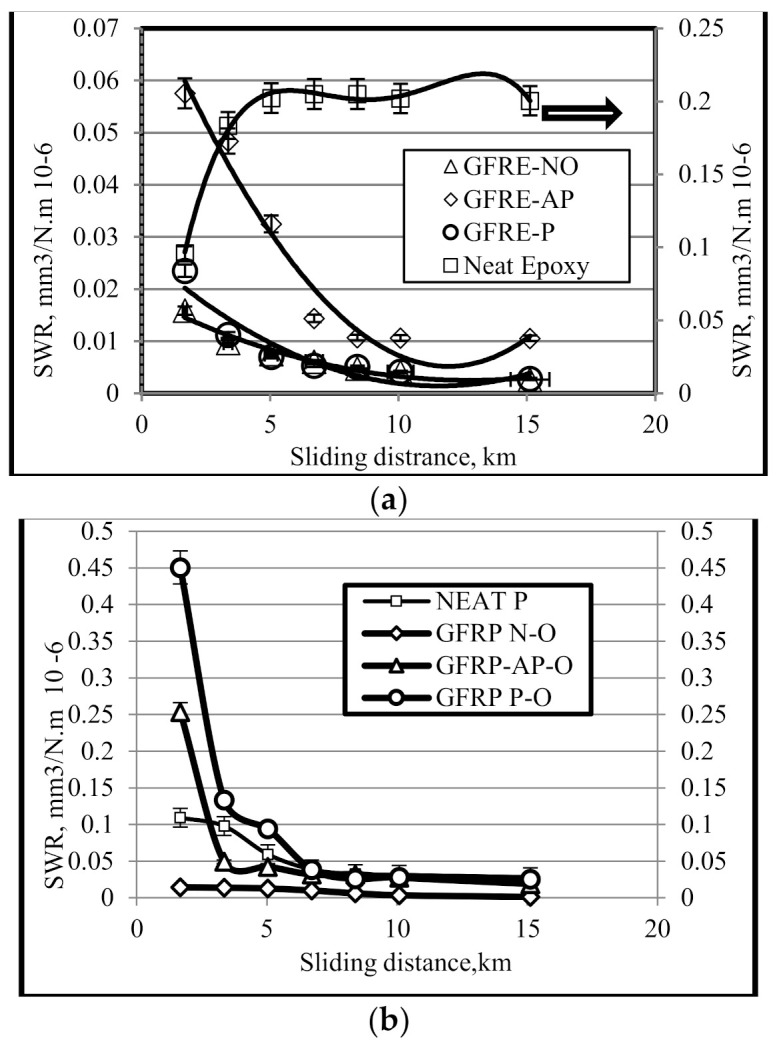
The variation in the SWR as a function of sliding distance for (**a**) NE and GFRE and (**b**) NP and GFRP.

**Figure 5 polymers-13-02230-f005:**
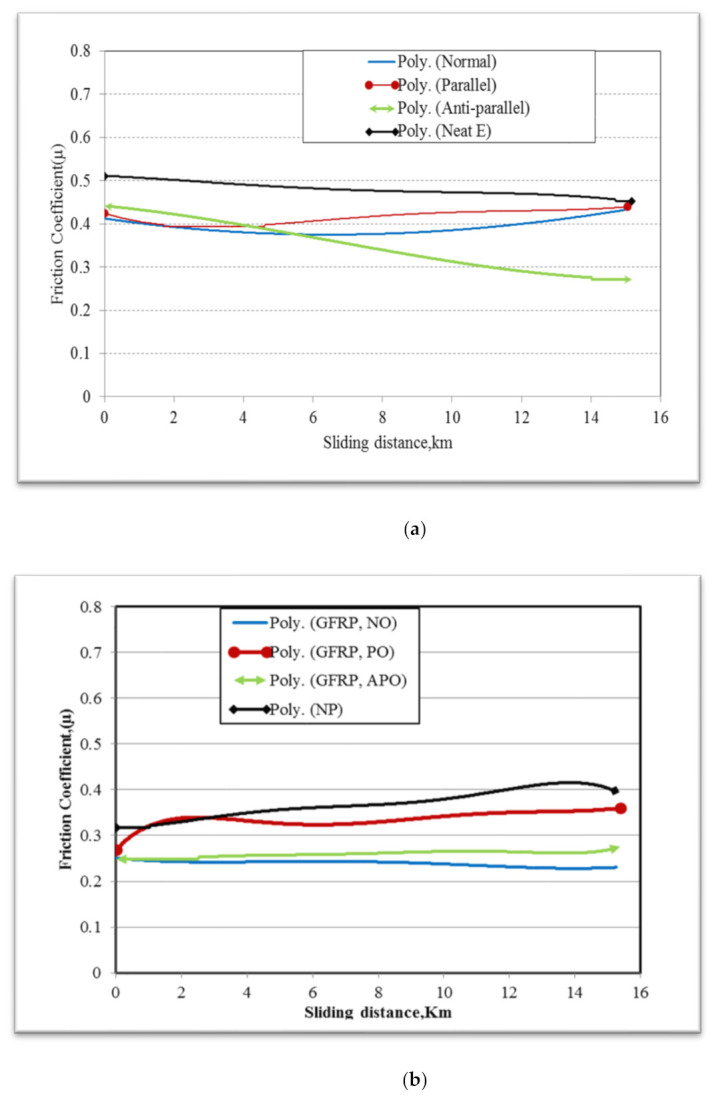
Friction coefficient of (**a**) GFRE and NE and (**b**) GFRP and NP.

**Figure 6 polymers-13-02230-f006:**
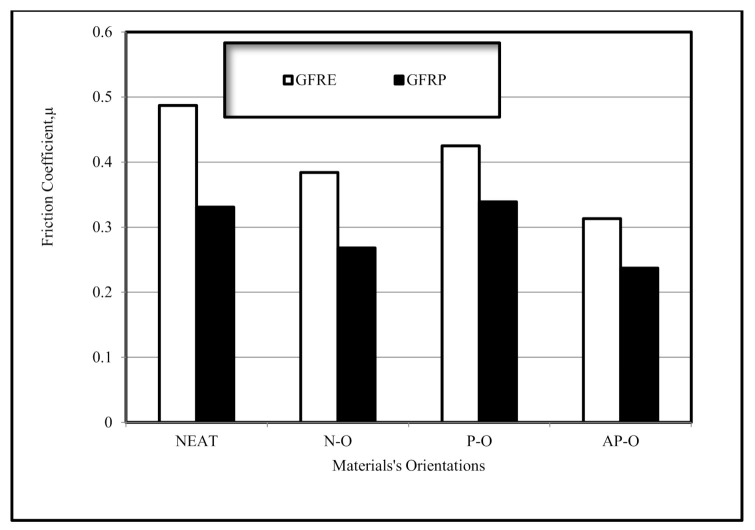
Summary of the friction coefficients of selected materials after reaching a steady state at 10 km sliding distance.

**Figure 7 polymers-13-02230-f007:**
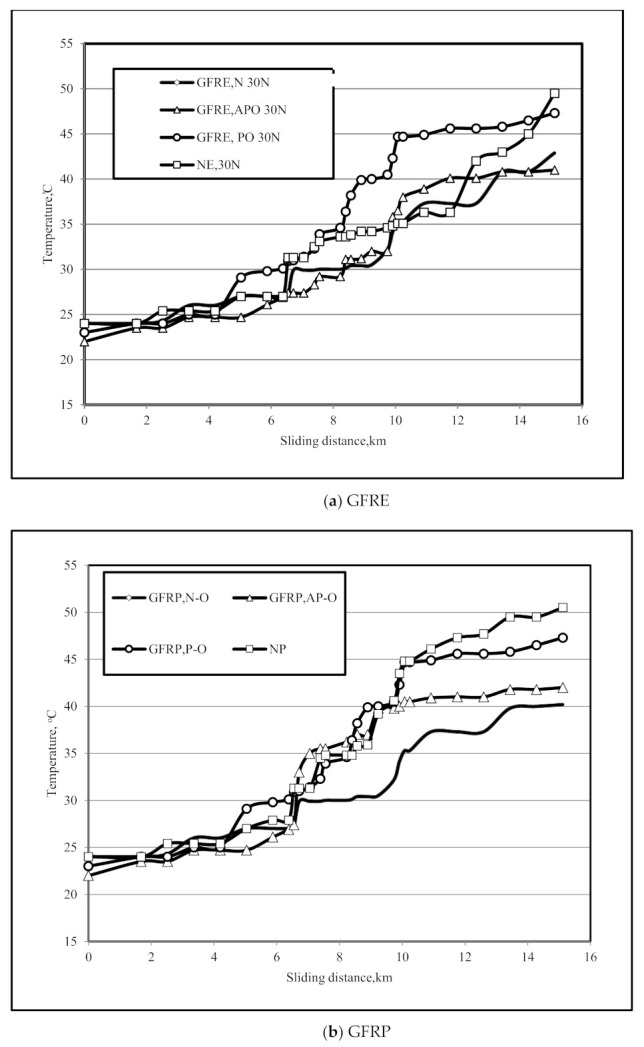
Interface temperatures of (**a**) GFRE and (**b**) GFRP under an applied load of 30 N.

**Figure 8 polymers-13-02230-f008:**
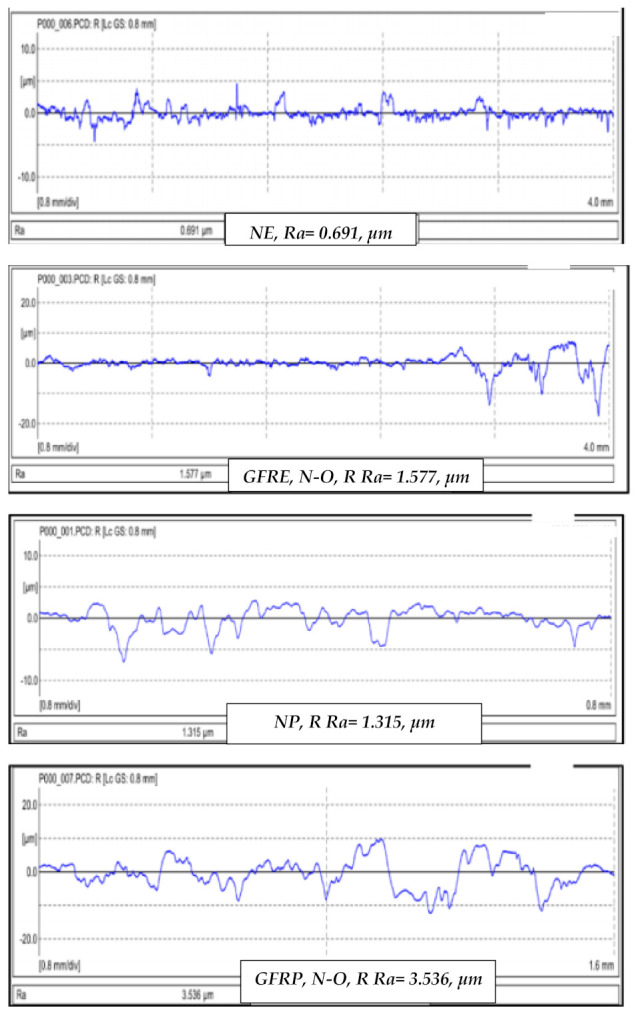
Sample roughness profiles Ra of selected materials at different operating parameters.

**Figure 9 polymers-13-02230-f009:**
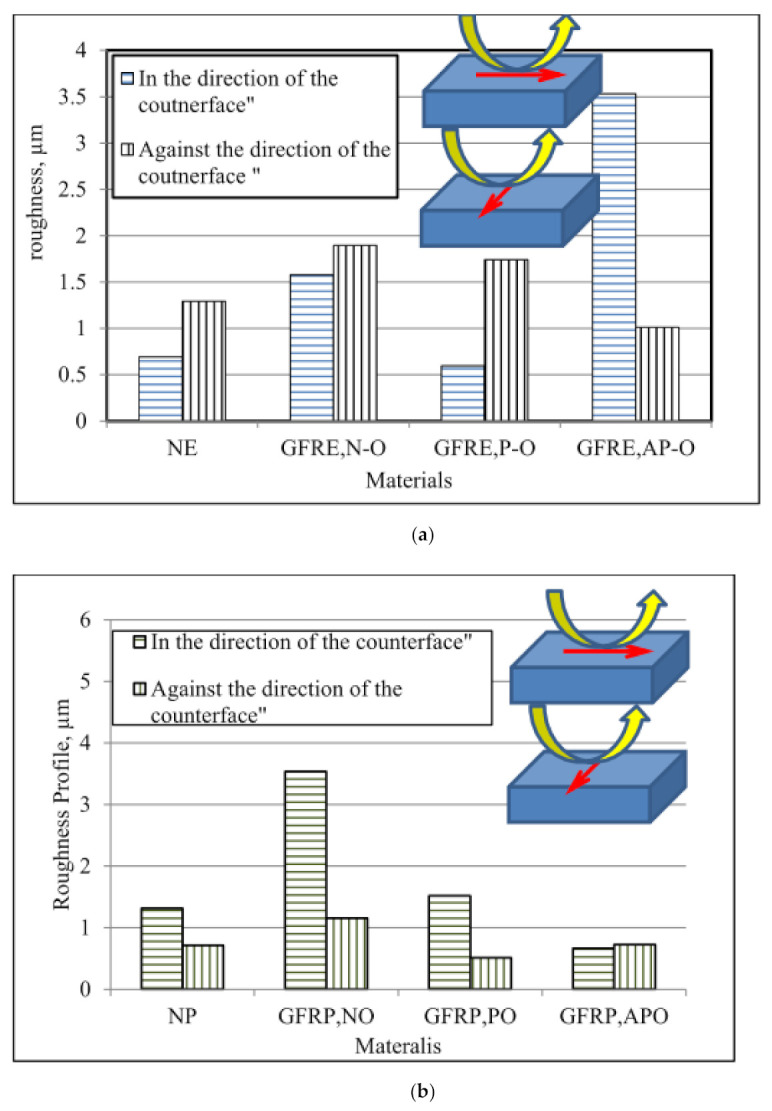
Roughness values of (**a**) GFRE and (**b**) GFRP under applied load of 30 N and sliding distance of 10–15 km.

**Figure 10 polymers-13-02230-f010:**
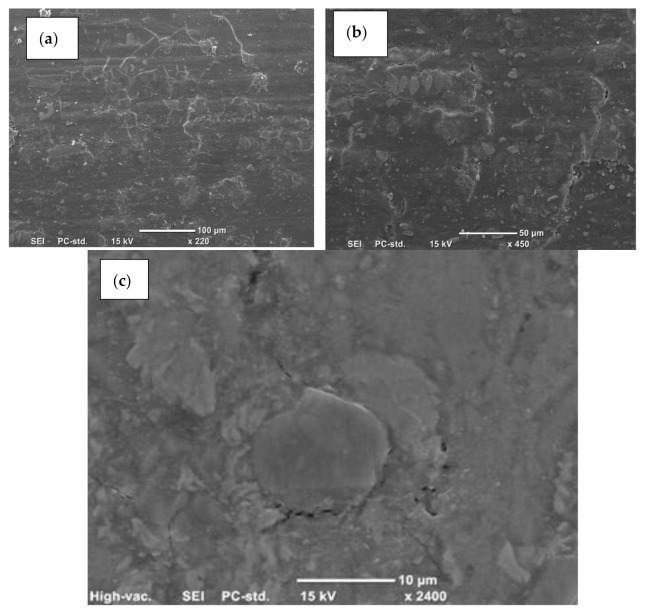
Micrographs of the worn surface of GFRE with N–O fibres following an applied load of 30 N.

**Figure 11 polymers-13-02230-f011:**
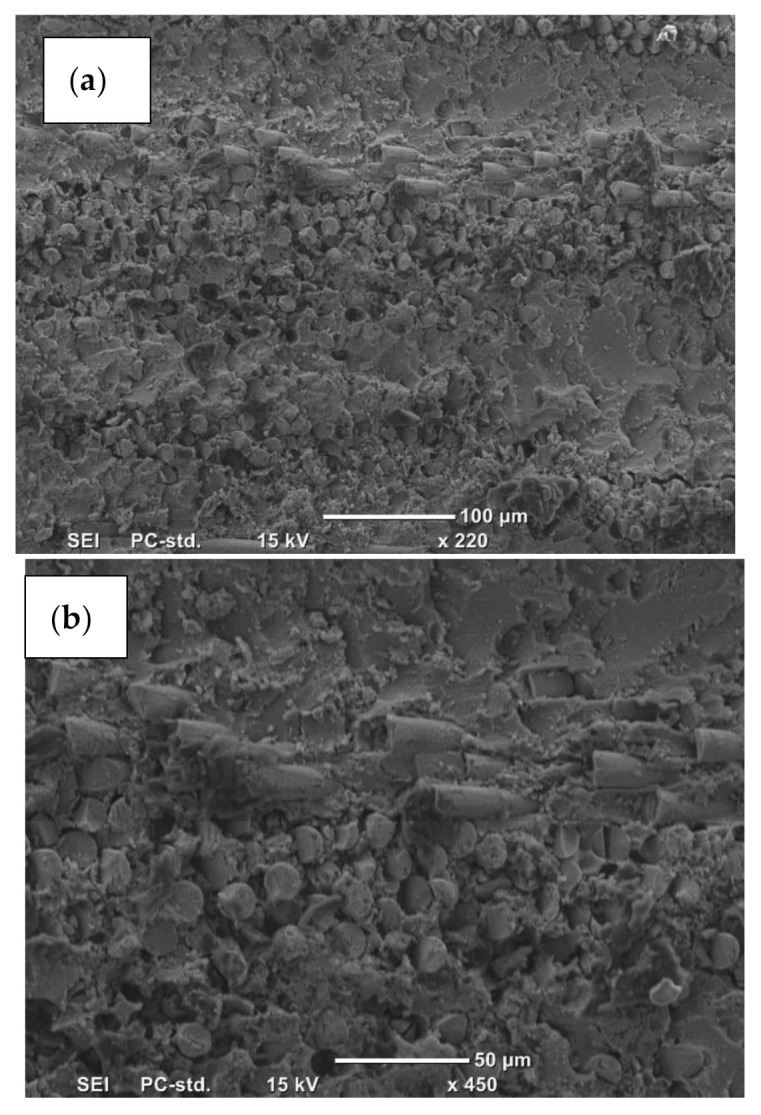
Micrographs of the worn surface of GFRP with AP–O fibres following an applied load of 30 N.

**Figure 12 polymers-13-02230-f012:**
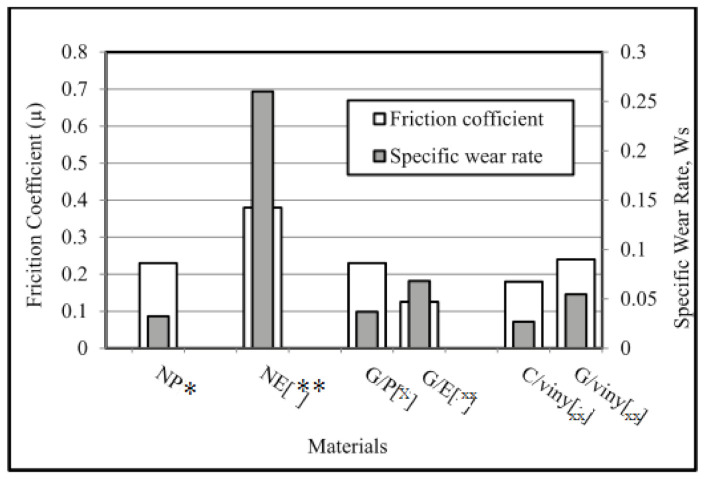
SWR and frictional behaviour of several composites, note that * is ref. No. [15], ** is ref. No. [18], x is ref. No. [1], and xx is ref. No. [3].

**Table 1 polymers-13-02230-t001:** Summary of the specific wear rate (SWR) of polyester and epoxy composites considering different orientations at applied load of 30 N after 10 km sliding distance.

Material	Orientation	SWR, mm^3^/N.m 10^−8^
Near Polyester		5 ± 0.0025
Neat Epoxy		5.5 ± 0.23
Polyester Composites	Parallel	3 ± 0.2
	Anti-Parallel	4.8 ± 0.26
	Normal	0.9 ± 0.19
Epoxy composites	Parallel	0.2 ± 0.18
	Anti-Parallel	1 ± 0.195
	Normal	0.2 ± 0.098

## Data Availability

Not applicable.

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
