# Peer review of "Epoxy and Polyester Composites’ Characteristics under Tribological Loading Conditions"

_polymers, 2021, doi:10.3390/polym13142230_

Round 1

Reviewer 1 Report

Dear Editor: I would like to express my deep thanks for inviting me to review the manuscript ID: polymers-1275152

Title:    Epoxy and polyester composites characteristics under tribological loading conditions

Authors: J.G. Alotaibi, Ayedh Eid Alajmi, G. A. Mehoub and B.F. Yousif

Comments:

Abstract:

Concise the abstract and rewrite according to the experimental results.

Introduction part:

  1. Explain in detail in introduction section why you select the following condition e.g., an applied load of 30 N and a sliding velocity of 15 2.8 m/s for specific applications.
  1. Please discuss in novelty and aim of this work.

Materials and Methods:

  1. Please mention how many samples used for wear test.
  2. Add all fibres manufacturer detail and fibres diameter.
  3. In equation 1, please mention all symbols, e.g., SWA= specific wear rate, etc.
  4. Explain in detail the in-situ temperature measurement system during wear teat that you represented in figure 8.

Results and discussion:

  1. Delete these sentences “The results for the GFRP and GFRE tribological tests are presented according to 127 various operating parameters. Frictional and wear parameters of the composites and thermoset include friction coefficients, interface temperatures and SWR. Surface morphology and roughness profile of the worn surfaces are also provided to help explain 130 the experimental wear and frictional results”.
  2. Figure 1 quality is very poor, replace it by high quality images that can clearly show the fibre/matrix interface.
  3. Explain in detail the variation of SWA of different polymer composite and correlate their worn surface.
  4. Please add standard deviation in Figure 3 &4.
  5. Figure 5 does not provide clear information, replace it by table.
  6. Figure 11 & Figure 12 quality are very poor, replace it by high quality images that can show the wear morphology and mechanism.

RECOMMENDATION

After reviewing the enclosed manuscript for “Polymers”, the present manuscript contains some kinds of scientific analysis but it is mandatory required to modify according to the preceding remarks. So, the manuscript can be accepted for publication after major revisions have been made.

Author Response

Epoxy and Polyester Composites Characteristics under Tribological Loading Conditions

Reviewer 1

The authors would like to thank the review for the constructive and valuable comments inn improving this article

Abstract:

Concise the abstract and rewrite according to the experimental results.

The abstract is rewritten and the experimental results are included as shown in the tracked version of the article

Introduction part:

  1. Explain in detail in introduction section why you select the following condition e.g., an applied load of 30 N and a sliding velocity of 2.8 m/s for specific applications.

In the revised version, this has been addressed in the introduction section as “In this work, the operating parameters were limited to 30 N of applied load and 2.8 m/s of sliding velocity. This is mainly due to the limitation of thermosets performance at severe tribological loading conditions (above 50 N applied load and higher than 2.8 m/s sliding velocity), [14 and 15]. In the literature, high applied load of 50 N and sliding velocity of 3.9 m/s, the surface of the thermosets dramatically softened and high removal of material have been reported leading to decomposition of the composites.

  1. Please discuss in novelty and aim of this work.

In the revised version, this has been addressed in the introduction section as “In the light of the above, the need to understand the influence of the fibre orientations on the tribological performance of both epoxy and polyester motivated the current study. The main aim of the work is identifying the optimum fibre orientation in both composites owing to gain the optimum performance of the composite under tribological loading conditions. Furthermore, this will assist the industries and the researchers in implementing such composites in tribological applications and give further understanding on the critical impact of the fibre orientation on the performance of such composites.

Materials and Methods:

  1. Please mention how many samples used for wear test.

This has been added in the revised version under experimental procedure section as “ Each set of test has been repeated three times for three different samples”.

  1. Add all fibres manufacturer detail and fibres diameter.

It is added as

GFRP can exhibit different lengths, widths and weights. In the specimen used in this study, fibre length was 20–30 mm, and fibre mass was 450 g/m2, fibre diameter is 10 µm -30 µm. Unsaturated polyester (Revesol P9509) was added to methylethylketone peroxide at ambient room temperature. Both reinforcement and polyester materials were supplied by Kong Tat Company (Malaysia).

Sorry I could not have the manufacturer of the glass fibres. but I know the supplier. 

  1. In equation 1, please mention all symbols, e.g., SWA= specific wear rate, etc.

It is given in the revise version as

Where

SWR: specific wear rate

L: applied load

  1. Explain in detail the in-situ temperature measurement system during wear teat that you represented in figure 8.

The below para is added in the revised version at the end of the exp. Procedure.

The experimental setup includes an infrared thermometer (Extech 42580) to deduct the interface temperature between the surface and the counterface. The calibration and the measurement procedure was detailed in [1]. In summary, a thermocouple was placed between the composite and the counterface surfaces at stationary position. The counterface was heated using external heat gun and then the interface temperature was captured by the thermocouple and the infrared simultaneously. A calibration equation was developed and used to determine the real interface temperature while using the infrared thermometer.

Results and discussion:

  1. Delete these sentences “The results for the GFRP and GFRE tribological tests are presented according to 127 various operating parameters. Frictional and wear parameters of the composites and thermoset include friction coefficients, interface temperatures and SWR. Surface morphology and roughness profile of the worn surfaces are also provided to help explain 130 the experimental wear and frictional results”.

Removed

  1. Figure 1 quality is very poor, replace it by high quality images that can clearly show the fibre/matrix interface.

The SEMs replaced with the new ones a copy below: 

Fig. 1 Micrographs of the original surfaces of (a) GFRP and (b) GFRE

  1. Explain in detail the variation of SWA of different polymer composite and correlate their worn surface.

This para is added in the revised version after figure 12.

The results reported in Fig. 5 showed that the epoxy composites performed better than the polyester composites at all of the fibers orientations. This can be due to many factors. Firstly, epoxy resin is much harder than the polyester and under tribological loading conditions, the wear performance improves with the higher hardness of the surface, [2]. On the other hand, the interaction between the glass fibers and the resin may differ based on the type of resin. Epoxy is well known to have very high adhesion properties compared to the polyester, [3].  This may suggest that the interfacial adhesion between the fibres and the resin may influence the material removal from the surface of the composite during the rubbing process. By comparing Figs. 11 and 12, it suggests that the epoxy regions on the worn surface were adhering well with the fibres and the ends of the fibers carried the load during the sliding. However, Fig. 12 b showed broken fibers and deterioration in the polyester region near the fibrous. This can explain the better performance of epoxy composites compared to the polyester ones.

  1. Please add standard deviation in Figure 3 &4.

Added in Figs. 4 and 5.

  1. Figure 5 does not provide clear information, replace it by table.

It is replaced and para to explain it is given as well. It is inserted in the revised version and copy is provided below

Table 1 summarises the SWRs of selected materials after reaching a steady state at 10 km. The optimum SWR for both GFRE and GFRP occurred in the N-O composites, which may be related to their mechanical properties in terms of interfacial adhesion and strength, lowering the hardness of the composite surface. For composites with AP-O fibres, the SWR of GFRP was higher than that of GFRE, while there was a marked difference between composites with P-O fibres. NE and NP had higher SWR values than either of the composites.

Table 1 summary of the specific wear rate (SWR) of polyester and epoxy composites considering different orientations at applied load of 30 N after 10 km sliding distance.

Material

Orientation

SWR, mm³/N.m 10  -8

Near Polyester 

5  ± 0.0025

Neat Epoxy

5.5  ± 0.23

Polyester Composites

Parallel

3  ± 0.2

Anti-Parallel

4.8  ± 0.26

Normal

0.9  ± 0.19

Epoxy composites

Parallel

0.2 ± 0.18

Anti-Parallel

1  ± 0.195

Normal

0.2  ± 0.098

  1. Figure 11 & Figure 12 quality are very poor, replace it by high quality images that can show the wear morphology and mechanism.

They are replaced by the below figures and explained as

In both composites, neat resins (Epoxy and Polyester) showed poor wear resistance, Table 1. With regards to the composites, when the fibres ends were exposed to the rubbing area in normal orientation (N-O), both composites performed better than other orientations (P-O and AP-O). the micrographs of the composites’ surfaces in N-O are presented in Figs. 10 and 11. The worn surface of the epoxy composites showed no remarkable damages on the surfaces expect a slight debonding associated with micro-cracks around the fibres. There is a clear deformation on the surface which can be due to film generation on the surface. There is no clear presence of fibres end which indicates that there is film generation on the composites surface. This may explain the smoothness of the surface when the roughness was measured and reported in Fig. 9. Furthermore, the deformation on the surface represents the high resistance of the surface for material removal owing to low specific wear rate, table 1. On the other hand, worn surface of the epoxy composites showed dramatical deterioration in the surface since there are fibre breakage, debonding, delamination and pull out of fibres as can be seen in Fig. 11. This leads to high roughness (Fig. 9) and high specific wear rate, table 1. Both figures (10 &11) supports the experimental results and the findings.

Fig. 10 Micrographs of the worn surface of GFRE with N-O fibres following an applied load of 30 N

Fig. 11 Micrographs of the worn surface of GFRP with N-O fibres following an applied load of 30 N

RECOMMENDATION

After reviewing the enclosed manuscript for “Polymers”, the present manuscript contains some kinds of scientific analysis but it is mandatory required to modify according to the preceding remarks. So, the manuscript can be accepted for publication after major revisions have been made.

All done with thanks

Reference (the numbers are different than the one in the revised version since it is auto update using Endnote)

  1. Chin, C.W. and B.F. Yousif, Potential of kenaf fibres as reinforcement for tribological applications. Wear, 2009. 267(9): p. 1550-1557.
  2. Pujar, V., et al., A review on mechanical and wear properties of fiber-reinforced thermoset composites with ceramic and lubricating fillers. Materials Today: Proceedings, 2021.
  3. Gopinath, A., M.S. Kumar, and A. Elayaperumal, Experimental investigations on mechanical properties of jute fiber reinforced composites with polyester and epoxy resin matrices. Procedia Engineering, 2014. 97: p. 2052-2063.

Reviewer 2 Report

Dear authors,

Your paper is an interesting experimental study reefer to the friction and wear of some composite materials.

To be accepted for publication I recommend  to consider following questions:

  1. In the experiments are indicated the normal force and sliding speed. Your testing specimens realize in contact with the disc a linear contact. In my opinion is important to determine the maximum Hertz contact as an important parameter. That means to use the Hertz theory and to know the Young modulus of the specimens.
  2. In the row 86 the "fibre mass was 450 g/m2 " . My question is if is correct this expression? I think that is a mistake!
  3.  In the rows 107-108 are indicated  an average roughness but roughness can be expressed by Ra, Rq, Rz , Rmax. Please indicate the type of the roughness parameter!
  4. In Eq. (1) please indicate the  the meaning of  the parameters: ΔW, ρ, L  and D and his units (mass, density, normal force and sliding distance).
  5. As a general observation ,please realize all the graphics  unitary as  font sizes!
  6. In Fig. 4 please indicate summary that are presented the variation of the SWR as function of sliding distance for.......
  7. Fig 5 is confusing, in left and in right are difference of units! What means left and right values? Also you are indicated only black and white color but way is gray color?
  8.  Please increase the fonts size in Fig. 6!
  9.  In Fig. 7 please indicate more clearer the difference between  GFRP and GFRE!
  10. Fig. 8 must be unitary: in a) are indicated the force 30N, in b) are not indicated 30N and not unity of temperature. Also, it is not clear from Fig. 8 b) if the black curve is for GFRP N-O!
  11. The roughness profiles are realized a lot of years ago ( 2008 and 2009). It may be a mistake. Also you are not indicated which are the roughness profile in the sliding direction and in opposite direction.
  12. If initial the contact is a linear one, during a time, as result of wear  must be a contact between a disc and a curve surfaces (generated by wear), with the modification of the contact pressure. Have you measured the final contact surface between specimens and the steel disk?
  13.  In Fig. 10  must indicated the type of roughness Ra, Rq ,Rz ! Also I consider that you  measured the roughness parameter in the sliding direction and in opposite direction and obtained  higher values in the sliding direction as result of adhesion and abrasion processes in the sliding direction. The word "coutner face" can create confusion.
  14. In Fig. 13 are compared your results  with other researchers results. The graphics must be realized with  indications of every reference more clearly. Is more confusion in this figure!

Author Response

Reviewer 2

The authors would like to thank the review for the constructive and valuable comments inn  improving this article

Dear authors,

Your paper is an interesting experimental study reefer to the friction and wear of some composite materials.

To be accepted for publication I recommend  to consider following questions:

  1. In the experiments are indicated the normal force and sliding speed. Your testing specimens realize in contact with the disc a linear contact. In my opinion is important to determine the maximum Hertz contact as an important parameter. That means to use the Hertz theory and to know the Young modulus of the specimens.

Thanks for the comments. In the revised version is written

The contact between the two surfaces is cylindrical against flat and the Maximum Hertzian contact pressure (MPa) is determined to be 3.2, 1.6, 10.5 and 6.8 for neat epoxy, neat polyester, GFRE, and GRFP, respectively.

  1. In the row 86 the "fibre mass was 450 g/m2 " . My question is if is correct this expression? I think that is a mistake!

Thanks for the note this information is provided by the supplier and it is common unit for the glass fiber mat as you can see in the link below

http://www.colan.com.au/compositereinforcement/non-woven/chopped-strand-mat/fibreglass-csm-emulsion-450g-m2-1040mm.html

  1.  In the rows 107-108 are indicated  an average roughness but roughness can be expressed by Ra, Rq, Rz , Rmax. Please indicate the type of the roughness parameter!

Thanks it is the Ra, and it is written in the revised version

  1. In Eq. (1) please indicate the  the meaning of  the parameters: ΔW, ρ, L  and D and his units (mass, density, normal force and sliding distance).

It is described in the revised version below the equation

  1. As a general observation ,please realize all the graphics  unitary as  font sizes!

It is corrected according to the format of the journal

  1. In Fig. 4 please indicate summary that are presented the variation of the SWR as function of sliding distance for.......

The caption changed to

  1. Fig 5 is confusing, in left and in right are difference of units! What means left and right values? Also you are indicated only black and white color but way is gray color?

It is replaced by table

Table 1 summary of the specific wear rate (SWR) of polyester and epoxy composites considering different orientations at applied load of 30 N after 10 km sliding distance.

Material

Orientation

SWR, mm³/N.m 10  -8

Near Polyester 

5  ± 0.0025

Neat Epoxy

5.5  ± 0.23

Polyester Composites

Parallel

3  ± 0.2

Anti-Parallel

4.8  ± 0.26

Normal

0.9  ± 0.19

Epoxy composites

Parallel

0.2 ± 0.18

Anti-Parallel

1  ± 0.195

Normal

0.2  ± 0.098

  1.  Please increase the fonts size in Fig. 6!

Done

  1.  In Fig. 7 please indicate more clearer the difference between  GFRP and GFRE!

It is cleared as below

Fig. 6 Summary of the friction coefficients of selected materials after reaching a steady state at 10 km sliding distance

  1. Fig. 8 must be unitary: in a) are indicated the force 30N, in b) are not indicated 30N and not unity of temperature. Also, it is not clear from Fig. 8 b) if the black curve is for GFRP N-O!

It is corrected in the revised version

  1. The roughness profiles are realized a lot of years ago ( 2008 and 2009). It may be a mistake. Also you are not indicated which are the roughness profile in the sliding direction and in opposite direction.

The date is wrong and this is forget when the machine used .. the date was not updated. In the revised version the date is removed. All the roughness profile were taken normal to the sliding direction.. this is explained in revised version the results part as

It should be mentioned here that the roughness profile was determine in normal direction to the sliding direction

  1. If initial the contact is a linear one, during a time, as result of wear  must be a contact between a disc and a curve surfaces (generated by wear), with the modification of the contact pressure. Have you measured the final contact surface between specimens and the steel disk?

Unfortunately, I did not. This is very important point and will be considered in the future work. Thanks

  1.  In Fig. 10  must indicated the type of roughness Ra, Rq ,Rz ! Also I consider that you  measured the roughness parameter in the sliding direction and in opposite direction and obtained  higher values in the sliding direction as result of adhesion and abrasion processes in the sliding direction. The word "coutner face" can create confusion.

In each profile Ra is indicated in the revised version also the title of the figure contain Rs as well.

This is written in the revised version “It should be mentioned here that the roughness profile was determine in normal direction to the sliding direction”

I check the article and made sure that Counterface word used as standard for the stainless steel disk.

  1. In Fig. 13 are compared your results  with other researchers results. The graphics must be realized with  indications of every reference more clearly. Is more confusion in this figure!

Unfortunately, it is a challenge to add the refences in the figure itself since we are using Endnote software and does not update the reference no. in the figure. In the revised version the caption of the figure become clear as written as

SWR and frictional behaviour of several composites, note that * is ref. No [15], ** is ref. No [12], x is ref. No [1], and xx is ref. No [10].

Round 2

Reviewer 1 Report

Authors addressed all of the review comments in the revised manuscript.